# Quercetin Synergistically Inhibit EBV-Associated Gastric Carcinoma with *Ganoderma lucidum* Extracts

**DOI:** 10.3390/molecules24213834

**Published:** 2019-10-24

**Authors:** Sora Huh, Seulki Lee, Su Jin Choi, Zhexue Wu, Jae-Han Cho, Lina Kim, Yu Su Shin, Byung Woog Kang, Jong Gwang Kim, Kwanghyeon Liu, Hyosun Cho, Hyojeung Kang

**Affiliations:** 1College of Pharmacy and Research Institute of Pharmaceutical Sciences, Kyungpook National University, Daegu 41566, Korea; hsl367@naver.com (S.H.); sujinchoi88@naver.com (S.J.C.); wuzhexue527@gmail.com (Z.W.); rlaflsk0424@hanmail.net (L.K.); dstlkh@knu.ac.kr (K.L.); 2College of Pharmacy and Innovative Drug Center, Duksung Women’s University, Seoul 01369, Korea; mchild1978@naver.com; 3Mushroom Research Division, National Institute of Horticultural & Herbal Science, Rural Development Administration, Eumseong 27709, Korea; limitcho@korea.kr; 4Department of Medical Crop Research, National Institute of Horticultural and Herbal Science, Rural Development Administration, Eumseong 27709, Korea; totoro69@korea.kr; 5Department of Oncology/Hematology, Kyungpook National University Hospital, Kyungpook National University School of Medicine, Kyungpook National University Cancer Research Institute, Kyungpook National University, Daegu 41566, Korea; bwkang@knu.ac.kr (B.W.K.); jkk21c@knu.ac.kr (J.G.K.)

**Keywords:** Quercetin, *Ganoderma lucidum* extracts, EBV-associated gastric carcinoma, Ganoderic acid A

## Abstract

Mycotherapy has been shown to improve the overall response rate during cancer treatment and reduce some chemotherapy-related adverse events. *Ganoderma lucidum* is a traditional mushroom used for pharmaceutical purposes. *G. lucidum* extracts (GLE) showed potential antitumor activities against several cancers. These tumor inhibitory effects of GLE were attributed to the suppression of the proliferation and metastasis of cancer cells. Epstein-Barr virus (EBV)-associated gastric carcinoma (EBVaGC) is defined as the monoclonal proliferation of carcinoma cells with latent EBV infection. The inhibitory effects of GLE against EBVaGC are questionable. The aim of this study was to investigate GLE as potential antitumor agents and a counterpart of quercetin (QCT) for the cotreatment in suppressing EBVaGC development. Therefore, this study conducted antitumor assays using a EBVaGC xenograft mice model and found that GLE could suppress tumor development. These inhibitory effects were significantly augmented by the low concentration of the quercetin (QCT) cotreatment in the xenograft mice. The addition of GLE in low concentrations synergistically reinforced QCT-induced apoptosis and EBV lytic reactivation. GLE contains various polysaccharides and triterpenes, such as ganoderic acid. Interestingly, the addition of ganoderic acid A (GAA) could produce similar bioactive effects like GLE in QCT-mediated antitumor activity. The GAA addition in low concentrations synergistically reinforced QCT-induced apoptosis and EBV lytic reactivation. GAA was sufficiently effective as much as GLE. Therefore, our results suggested that QCT-supplemented GLE could be a potential food adjunct for the prevention of EBVaGC development.

## 1. Introduction

*Ganoderma lucidum* Karst is a wood-rotting mushroom generally growing on tree stumps. The fruiting bodies of *G. lucidum* Karst are widely used in Northeast Asia as a natural product to treat chronic hepatitis, nephritis, hepatopathy, gastric ulcer and insomnia. Additionally, *G. lucidum* extracts (GLE) have been shown to exhibit anticancer properties by inhibiting tumor growth, angiogenesis, metastasis and telomerase activity [1,2,3,4]. Furthermore, GLE exhibited antiviral activities during various viral lifecycle stages, including viral attachment to host cell receptors and viral penetration into host cells. For example, acidic protein bound polysaccharide (APBP) was isolated from water-soluble GLE and showed direct virucidal effects against Herpes simplex virus (HSV)-1 and HSV-2 [5]. APBP inhibited up to 50% of the attachment of HSV-1 and HSV-2 to Vero cells and prevented both viruses from penetrating Vero cells. The antiherpetic activity of APBP suggested that APBP could impede the complex interactions of viruses with cell membranes. 

The bioactive ingredients of GLE include triterpenes, steroids, polysaccharides and proteoglycans. More than 100 GLE triterpenes have been isolated and designated as ganoderic acid A-T, lucidenic acid A-P, lucidenolactone, methyl ganoderate E-F, and others [6]. These triterpenes displayed several biological activities, including antitumor activity [1,7]. Ganoderic acid A had significant inhibitory effects against both Epstein-Barr virus (EBV) early antigen (EA) and capsid antigen (CA) activation at 16 nmol [4]. Ganoderic acid B showed significant anti-human immunodeficiency virus (anti-HIV)-1 protease activity with IC_50_ values of 20–90 μM [8]. Lucidenic acids showed inhibitory effects against 12-*O*-tetradecanoylphorbol-13-acetate (TPA)-induced EBV EA activation in Raji cells (96–100% inhibition at 1 × 10^3^ mol ratio/TPA) [2]. Lucidenic lactone inhibited HIV-1 reverse transcriptase activity [9]. 

EBV is a human herpes virus that causes infectious mononucleosis and is associated with both B cell and epithelial cell malignancies [10]. EBV can cause two types of infections in cells, latent and lytic [11]. In the latent state, EBV genome exists in the nucleus as an episome, which is chromatinized with histones and expresses only a few latent genes. As only some gene products that can be targeted by the host immune system are expressed, EBV can escape a host immune attack and indefinitely survive in the host. The stimulation of EBV lytic cascade using chemical or biological agents first leads to the expression of two viral immediately early (IE) genes, *BZLF1* and *BRLF1*. These IE genes encode transcriptional activators and coordinately induce the expression of early (E) genes, such as *BALF5* (DNA polymerase catalytic subunit) and *BALF2* (single-stranded DNA-binding protein). By activating these EBV enzymes and proteins, EBV potently replicates its genome in a rolling circular manner. 

EBV-associated epithelial cancers (EBVaECs) have received considerable interest for the past two decades because they represent 80% of all EBV-associated malignancies [12]. Nasopharyngeal carcinomas (NPC) and EBV-associated gastric carcinomas (EBVaGCs) are the most common EBVaECs with 78,000 and 84,000 of new cases, respectively, annually worldwide [12]. EBVaGCs represent approximately 10% of all gastric cancers. Among EBVaGCs, 16% are conventional gastric adenocarcinomas and 89% are lymphoepithelioma-like gastric carcinomas (LELC) [13]. LELC is a poorly differentiated carcinoma with dense lymphocytic infiltration. The selective expression of EBV genes (type I latency) contributes to the malignant transformation of epithelial cells by disrupting cellular processes and signaling pathways. EBVaGC genome contains distinct mutations and methylation patterns, indicating that EBV stimulates a unique and alternate tumorigenesis pattern in EBVaGCs [14]. 

Flavonoids represent a class of secondary metabolites present in plants and fungi. Over 5000 naturally occurring flavonoids from various plants have been discovered and structurally characterized. They are classified according to their chemical structure as follows: anthoxanthins, flavanones, flavanonols, flavans, and anthocyanidins [15]. Quercetin (QCT) and isoliquiritigenin (ISL) are present in several herbs and belong to anthoxanthins and flavanones, respectively. QCT has shown anticancer effects owing to its antioxidant activity, the inhibition of enzymes that activate carcinogens, the modification of signal transduction pathways and the interactions with receptors and other proteins [16]. In fact, QCT contains a polyphenolic structure that acts as a scavenger of free radicals, and is therefore, responsible for its antioxidant effects. In addition, QCT acts as an agonist of the G protein-coupled estrogen receptor that plays a role in breast cancer progression and tamoxifen resistance [17]. These findings suggest that QCT is a multifunctional compound with potential anticancer properties.

Regarding its anti-EBV and antitumor effects, QCT has been shown to be cytotoxic to SNU719 EBVaGC cells. Additionally, QCT induced apoptosis and a strong EBV lytic reactivation in SUN719 cells, suggesting that QCT could be a promising candidate anti-EBV and antitumor agent [18]. Triterpenoids, such as ganoderic acid A (GAA) and ganoderic acid B found in GLE, were shown to moderately inhibit the activity of telomerase at 10 μM [4]. Molecular docking showed that GAA could inhibit telomerase as a ligand. These results suggested that triterpenoids could exhibit antitumor effects in EBVaGC cells. GLE and QCT might exert significant synergistic anti-EBV and antitumor activities against EBVaGC cells. This study aimed to elucidate the synergistic effects of GLE and QCT cotreatment on inhibiting EBV infection and EBVaGC development. This study showed that cotreatment with GLE and QCT at low concentrations could synergistically enhance apoptosis and induce strong EBV lytic reactivation in EBVaGC cells.

## 2. Results

### 2.1. Cotreatment of GLE and QCT Produced Synergistic Cytotoxic Effects 

Triterpenoids isolated from GLE could inhibit telomerases and were reported to have inhibitory effects on the activation of EBV early antigen (EA) and capsid antigen (CA). These inhibitory effects of triterpenoids appeared only on EBV-associated nasopharyngeal carcinoma (EBVaNPC) but not on Raji and B95-8 cells. Thus, triterpenoids were argued to be useful of preventing EBVaNPC [4]. QCT was reported to eliminate DNMT1 and DNMT3A in EBVaGC and caused demethylation on EBV lytic promoters. These epigenetic effects of QCT could induce strong EBV lytic reactivation [18]. In this study, it was questioned whether the cotreatment of GLE and QCT could make any different bioactivity in EBVaGC. To this aim, cytotoxicity assays were first conducted and the results were subsequently subjected to the Chou-Talalay analysis [19]. The combinations for the cotreatment of GLE and QCT were prepared in a series of concentrations and evaluated for their cytotoxicity against SNU719 cells. The combination index (CI) was calculated with relative cytotoxicity by serial combinations of GLE and QCT. Subsequently, Fa-CI and median-effect plots were generated. The Fa-CI plot of GLE and QCT cotreatments indicated that the cotreatment of GLE and QCT worked synergistically in high concentrations of QCT (Figure 1A). Most of the CI values in QCT low concentrations were lower than 1, suggesting synergistic cytotoxic effects against SNU719 cells by the low concentrations of QCT and GEL cotreatment (0.24 ~ 7.75 μM). In addition, the median-effect plot of GLE and QCT cotreatments supported the Fa-CI plot analysis (Figure 1B). Most median-effect equation values (Fa/Fu) of the cotreatments were located within the slopes of GLE and QCT single treatments, supporting the synergistic cytotoxic effects of GLE and QCT cotreatments. Taken together, all these data indicated that low concentrations of GLE and QCT cotreatment were likely to produce the cytotoxic synergism against SNU719 cells. 

### 2.2. Cotreatment of GLE and QCT at High Concentration did not Synergistically Cause Anti-Tumor Effect 

As combination of GLE and QCT in low concentrations showed stable synergisms in causing cytotoxicity against EBVaGC, SNU719 cells. It was questioned if the cotreatment with GLE and QCT could exert synergistic effects in suppressing tumor development. To test this hypothesis, athymic nude mice were used. This is a xenograft mouse model that does not reject human cells when human tumor cells are transplanted under the skin as previously described [20]. MKN1-EBV cells were injected into nude mice and GLE, QCT and isoliquiritigenin (ISL) (30 mg/kg) were orally administered as either a single treatment or cotreatment (Figure 2A). At twenty-three days post tumor injection, tumor development was observed (Figure 2B) and tumor sizes on the thighs of xenograft nude mice were measured (Figure 2C,D). The single treatment with GLE, QCT and ISL resulted in significant suppression of tumor development. Tumors in mice treated with a single agent developed marginally, up to 300 mm^3^, whereas tumors in the control group developed maximally, up to 700 mm^3^. Moreover, the cotreatment with GLE, QCT and ISL showed a remarkable delay in tumor development. The tumors in mice receiving the cotreatment were poorly developed up to 250 mm^3^. However, no significant difference in MKN1-EBV tumor size was observed between the tumors of mice treated with single treatment (30 mg/kg) and cotreatment (30 mg/kg each). Furthermore, no statistically significant difference could be found between any two groups: control versus QCT (or ISL), control versus GLE, control versus GLE + QCT (or ISL), QCT (or ISL) versus GLE, QCT (or ISL) versus GLE + QCT (or ISL) and GLE versus GLE + QCT (or ISL). These results indicated that the cotreatment with GLE and either QCT or ISL at a high concentration was unlikely to synergistically suppress EBVaGC development. 

### 2.3. Cotreatment of GLE and QCT at Low Concentration Synergistically Caused Anti-Tumor Effect

The cotreatment with GLE and QCT at 30 mg/kg resulted in a slight nonsignificant decrease in tumor size, compared to that of a single GLE treatment at 30 mg/kg. Thus, this study could not exclude the possibility that the cotreatment at 30 mg/kg each was too cytotoxic to produce significant synergistic antitumor effects. Therefore, lower concentrations of the compounds were used. GLE, QCT and ISL (10 mg/kg) were orally administered to nude mice either as a single treatment or cotreatment (Figure 3A). At twenty-three days post tumor injection, tumor development was observed (Figure 3B) and tumor sizes on the thighs of xenograft nude mice were measured (Figure 3C,D). A QCT and ISL single treatment at 10 mg/kg resulted in a significant delay in tumor development. The tumors in mice treated with a QCT or ISL single treatment marginally developed up to 700 mm^3^ and 600 mm^3^, respectively. However, the tumors in GLE alone-treated and control mice maximally developed up to 1200 mm^3^. Interestingly, the cotreatment with GLE and QCT at 10 mg/kg showed significant suppression of tumor development, compared to that of either a GLE or QCT single treatment. The GLE and QCT cotreatment suppressed tumor development up to 400 mm^3^. Statistically, two groups, such as the control versus GLE + QCT and GLE versus GLE + QCT, were significantly different in tumor sizes. In contrast, the cotreatment with GLE and ISL at 10 mg/kg did not show any significant difference in suppressing tumor development, compared to that of either GLE or ISL single treatment. Only the control versus GLE + ISL group was statistically significant in tumor sizes. These results indicated that the cotreatment with GLE and QCT at a low concentration might produce synergistic antitumor effects against EBVaGC development. 

### 2.4. Cotreatment of GLE and QCT Reduced Cell Viability and Induced Apoptosis 

Since the cotreatment with GLE and QCT at a low concentration produced synergistic antitumor effects, the potential underlying molecular mechanisms were further investigated. Therefore, the 50% cytotoxic dose (CD_50_) of GLE in SNU719 cells was determined to be 0.0532 mg/mL (Figure 4A). Then, a cell viability test was carried out to assess whether the cotreatment with GLE and QCT might exert synergistic cytotoxic effects in SNU719 cells. The addition of GLE (0.0133 mg/mL) to QCT at a series of increasing concentrations induced a significant increase in cell death, compared to that induced by QCT alone (Figure 4B). The cotreatment with 0.96 μM QCT and 0.0133 mg/mL GLE induced a 1.69-fold increase in cell deaths, compared to that induced by 0.96 μM QCT single treatment. These results suggested that QCT and GLE might have synergism in inducing cytotoxic effects at a low concentration. Since the QCT addition slightly enhanced the cytotoxicity of GLE in SNU719 cells, this study next investigated whether cytotoxicity was associated with apoptosis in SNU719 cells. A western blot assay was conducted with whole proteins of SNU719 cells. It was found from this analysis that the addition of GLE (0.0133 mg/mL) to QCT at a series of increasing concentrations enhanced QCT-induced apoptosis by increasing PARP1 cleavage, upregulating *CASP3* (caspase 3) and *CYCS* (cytochrome C) and downregulating *Bcl-2* (Figure 4C). To further verify the induction of PARP1 cleavage GLE and QCT cotreatment, nuclear proteins were separated from cytoplasmic proteins and a western blot assay was conducted. The GLE addition enhanced QCT-induced PARP1 cleavage in a concentration-dependent manner. The addition of GLE (0.0133 mg/mL) to QCT at a series of increasing concentrations remarkably increased the cleavage of PARP1 proteins, compared to that induced by QCT alone (Figure 4D). These results indicated that GLE might support the QCT-mediated cytotoxicity by enhancing apoptosis. 

### 2.5. Cotreatment of GLE and QCT Induced EBV Lytic Reactivation

Apoptosis can be activated through EBV lytic reactivation [21,22]. BZLF1 indirectly induces cell death via the inhibition of nuclear factor kappa-B (NF-κB) family protein, p65 [21] and the downregulation of CD74 [22]. Thus, this study further investigated whether the enhancement of QCT-mediated apoptosis by GLE was associated with EBV lytic reactivation. Therefore, a luciferase assay was conducted. The luciferase reporter construct contained a Renilla luciferase gene transcribed under the promoter of EBV lytic gene, *BHLF1* [23]. First, SNU719-BHLF1 luciferase cells were treated with QCT at different concentrations and 0.0133 mg/mL GLE. The GLE addition to QCT significantly enhanced the QCT-mediated BHLF1 expression in luciferase activity at low QCT concentrations (Figure 5A). The cotreatment with 0.96 μM QCT and 0.0133 mg/mL GLE enhanced luciferase activity up to 1.9-fold, compared to that of 0.96 μM QCT single treatment. These results indicated that GLE might be more effective in inducing EBV lytic reactivation than QCT. Since the cotreatment with GLE and QCT increased the activity of EBV lytic gene promoter, this cotreatment was investigated as to whether it could upregulate the EBV lytic gene expression to induce EBV lytic reactivation. Therefore, a qRT-PCR assay was conducted to determine the mRNA expression of EBV genes after the cotreatment with GLE and QCT. First, SNU719 cells were treated with QCT at different concentrations and 0.0133 mg/mL GLE. The overall expression level of EBV latent and lytic genes significantly increased after the cotreatment with GLE and QCT at a low concentration (Figure 5B). Interestingly, the mRNA expression of *EBNA1*, *LMP1*, *EBER*, and *BRLF1* reached the maximum after the addition of GLE at 0.0133 mg/mL and were decreasing with the increase in the QCT concentration (Figure 5B). These results suggested that both QCT and GLE might be similarly effective in upregulating the EBV gene expression. 

### 2.6. Quantification of GAA in GLE 

As mentioned above, GLE contained GAA and GAF as major bioactive compounds [24]. A previous study showed that GLE contained up to 0.43 mg GAA per 1 gm of fruiting body of *G. lucidum* strain KMCC 02,955 [25]. GAA might be considered a marker for selecting high-quality fruiting bodies for cancer treatment. Besides the antitumor effects, GAA exhibits strong antiviral effects against EBV [4] and HIV [8]. Similar to previous studies [25], a LC-MS/MS analysis was conducted to quantify GAA in GLE samples used in this study. The LC-MS/MS assay showed a clear peak of GAA at 3.63 min retention time (pressure = 183 bar) on the LC-MS/MS profile (Figure 6A). Then, the GLE samples were extracted independently using 99% ethanol and subjected to a LC-MS/MS analysis to quantify GAA in each GLE samples. It was found that all GLE samples contained GAA whose concentrations were from 2.04 ng/mg to 5.58 ng/mg (Figure 6B). This result suggested that some bioactivities of the GLE samples might come from GAA activity. 

### 2.7. Cotreatment of GAA and QCT Resembled that of GLE and QCT 

It was reasonable to define whether GAA bioactivity is similar to part of GLE bioactivities. The cotreatment with GAA and QCT was actually expected to produce potential synergistic antiviral and antitumor effects. To test this hypothesis, first, the CD_50_ value of GAA in SNU719 cells was determined to be 1.96 mM (Figure 7A). Then, a cell viability test was carried out to investigate whether the cotreatment with GAA and QCT could exert synergistic cytotoxic effects in SNU719 cells. The 0.49 mM GAA addition to QCT induced a significant increase in cell death by 2.73-fold, compared to that induced by QCT alone (Figure 7B). Since the GAA addition significantly enhanced the QCT-mediated cytotoxicity in SNU719 cells, this study next investigated whether the cytotoxicity was associated with apoptosis in SNU719 cells using Muse™ Annexin V & Dead Cell Kit as recommended by the manufacturer. Early apoptosis was more induced by the QCT and GAA cotreatment than the additive induction of both QCT and GAA single treatments, suggesting a synergistical effect on early apoptosis (Figure 7C). However, the QCT and GAA cotreatment did not synergistically induce late apoptosis and the cotreatment slightly enhanced late apoptosis than either the QCT or GAA single treatment (Figure 7C). As the addition of GLE enhanced QCT-mediated EBV lytic reactivation, it was similarly questioned whether a GAA addition could also enhance the QCT-mediated EBV lytic reactivation. Thus, a luciferase assay was conducted as previously conducted. First, SNU719-BHLF1 luciferase cells were treated with QCT at different concentrations with 0.49 mM GAA. The luciferase assay was then conducted 48 h post treatment. The 0.49 mM GAA addition to QCT increased luciferase activity up to 1.83-fold, compared to that induced by QCT alone (Figure 7D). Since the GAA addition increased the activity of the EBV lytic gene promoter, this study further verified whether the GAA addition could upregulate the EBV lytic gene expression to induce the EBV lytic reactivation. Thus, a qRT-PCR assay was conducted to determine the expression of EBV genes in SNU719 cells cotreated with QCT at different concentrations and GAA. The 0.49 mM GAA addition to QCT resulted in significant upregulations of *EBNA1* and *BZLF1* in a QCT concentration-dependent manner (Figure 7E). In particular, the GAA addition increased the *BZLF1* expression by 1,201-fold, compared to that of QCT alone. Taken together, these results suggested that, like GLE, GAA also is effectively working with QCT in inducing apoptosis, EBV lytic reactivation and EBV gene expression. 

## 3. Discussion

This study showed that the cotreatment of *G. lucidum* extracts (GLE) and quercetin (QCT) exerted synergistic antitumor and antiviral activities against SNU719 EBVaGC cells. Low concentrations of the cotreatment resulted in both the synergistic activation of cytotoxicity in SNU719 cells and the suppression of tumor development in MKN1-EBV xenograft mice. The addition of a low concentration GLE to QCT reinforced QCT-mediated cytotoxicity and QCT-mediated apoptosis in SNU719 cells. Furthermore, the addition also activated the EBV lytic gene promoter and upregulated EBV genes. Interestingly, ganoderic acid A (GAA), an effective molecule of GLE also showed similar bioactive features with QCT like GLE. The low concentrations of the GAA and QCT cotreatment reinforced QCT-mediated cytotoxicity and QCT-mediated apoptosis in SNU719 cells. Furthermore, the cotreatment activated the EBV lytic gene promoter and upregulated EBV genes, like GLE. 

QCT has been shown to exhibit potential anticancer properties owing to its antiproliferative, growth factor suppressive and antioxidant effects [26]. In addition, QCT can induce apoptosis, where it has been shown to reduce the growth of tumors and inhibit the spread of malignant cells. Moreover, it inhibited chemical carcinogen-induced cell transformation, which was evident as cell viability decreased, ROS generation and microRNA-21 elevation occurred in cancer cells [27]. Besides its antitumor effects, QCT has been shown to have antiviral activity through its anti-infective and anti-replicative abilities. The replication of herpes viruses, including EBV, was found to be affected by QCT [18]. EBVaGC exhibits mixed characteristics of both gastric carcinoma and EBV. Apoptosis and EBV lytic reactivation can control the fate of EBVaGC cells, where antitumor agents can induce apoptosis and antiviral agents can stimulate EBV lytic reactivation to further aggravate cell lysis. QCT exhibits both antitumor and antiviral activities against EBVaGC [18]. This dual function of QCT is likely to be synergistic. GLE, containing polysaccharides and triterpenes, is known to suppress the proliferation and metastatic potential of breast cancer cells by inhibiting Akt, AP-1 and NF-κB [28]. Moreover, GLE modulated the estrogen receptor signaling and inhibited oxidative stress-induced invasiveness of breast cancer cells [29]. Consistent with the results of previous studies on breast cancer, our study showed that GLE enhanced both QCT-mediated apoptosis and EBV-lytic reactivation. The effects of GLE were evident at low concentrations of QCT. Furthermore, QCT also enhanced the GLE-mediated upregulation of the EBV gene expression. Similarly, the effects of QCT were evident at low concentrations of GLE. Therefore, these results suggested that GLE enhanced QCT-mediated apoptosis, whereas QCT increased GLE-mediated EBV lytic reactivation. 

Mycotherapy has several benefits [30], where it improves the overall response rate during cancer treatment, enhances immunity owing to the stimulation of T cell proliferation and reduces some chemotherapy-associated adverse events, such as nausea and insomnia. Medicinal mushrooms have mainly been used in Asian countries for hundreds of years for the treatment of infectious diseases. More recently, they have been used for cancer treatment as adjuncts. Importantly, they have an extensive clinical history of safe use as single agents or in combination with chemotherapy. GLE has been intensively combined with mycotherapy for the treatment of broad-spectrum cancers [30,31]. GLE has shown antitumor activities against breast cancer, bladder cancer, prostate cancer, colorectal cancer and others. These tumor inhibitory effects of GLE were attributed to the disruption of the proliferation and metastasis of cancer cells. However, the fact that EBV is the initiator of gastric carcinoma should be considered in the treatment of EBVaGC. EBV should be eliminated to reduce the risk of EBVaGC relapse. In this study, QCT strongly induced EBV lytic reactivation. The synergistic effects of the GLE and QCT cotreatment might enhance QCT-mediated apoptosis and GLE-mediated EBV lytic reactivation. Therefore, QCT-supplemented GLE might be used as a medicinal food with a dual function as an antitumor and antiviral agent for EBVaGC treatment. 

To the author’s knowledge, this is the first study to report that QCT-supplemented GLE might be effective in treating EBVaGC. The synergistic antitumor and antiviral effects were supported by both in vivo and in vitro studies. Hence, the combined use of GLE and QCT could be beneficial in cancer treatment by alleviating the toxicity of conventional chemotherapy and improving the immune function. However, more studies are still needed to further elucidate the molecular mechanisms of the direct antitumor and antiviral effects of QCT-supplemented GLE. 

## 4. Materials and Methods

### 4.1. Preparation of GLE

*G. lucidum* strain CM980736, one of the mushroom culture collections of Chungnam-do Agricultural Technology Institute (Yesan, Korea) was grown into *G. lucidum* fruiting bodies by Yeonchen Cheongsan Mushroom Farming Association (YCMFA, Yeonchen, Korea). The *G. lucidum* fruiting bodies were obtained from YCMFA who prepared GLE for this study. Small molecules were extracted from 5.0 kg of *G. lucidum* fruiting bodies for 24 h at 25 °C using volume of 99% ethanol. This extract was repeated three times with the *G. lucidum* fruiting bodies previously extracted. The resultant primary extracts were filtered, concentrated, sterilized and freeze-dried. The final extract was approximately 0.5 g and named as *G. lucidum* extracts (GLE). All steps for GEL preparation were described previously [32]. GLE was dissolved in sterile distilled water to prepare a GLE stock solution (8 mg/mL) and stored at −20 °C until use. GAA was purchased from ChemFaces (Wuhan, China), dissolved in dimethyl sulfoxide (DMSO) to prepare a GAA stock solution (100 mM) and stored at −20 °C until use.

### 4.2. Cell Lines and Reagents

Both gastric carcinoma cell lines, SNU719 (EBVaGC) and MKN1-EBV (EBVaGC), were cultured in Roswell Park Memorial Institute (RPMI) 1640 medium (Hyclone, MA, USA) supplemented with 10% fetal bovine serum (Hyclone, MA, USA), antibiotics/antimycotics (Gibco, MD, USA), GlutaMAX^®^ (Gibco, MD, USA), and 25 mmol/mL HEPES (Sigma-Aldrich, MO, USA) at 37 °C in a 5% CO_2_ and 95% humidity incubator.

### 4.3. Ethics Statement

The animal experiments were conducted in accordance with the National Research Council’s Guide (IACUC, Seoul, Korea) for the Care and Use of Laboratory Animals. The experimental protocol was approved by the Animal Experiments Committee of Duksung Women’s University (permit number: 2014-016-007).

### 4.4. Antitumor Assay in a Xenograft Mouse Model

Nude mice (female, 5-week-old; Raonbio Co., Ltd., Seoul, Korea) were used as xenograft animal models for the assessment of the antitumor effects. The mice were individually housed in a pathogen-free controlled environment (23–27 °C under a 12 h light/12 h dark cycle) and provided with food and water ad libitum. The mice were first divided randomly into 2 groups (*n* = 30) and 5 × 10^6^ MKN1-EBV cells were then subcutaneously implanted into the dorsum next to the right hind leg of the mice in both groups. After 14 days, one group was used to test the synergistic antitumor effects of the cotreatment with a relatively high-concentration GLE and QCT or ISL, whereas the other group was used to test the synergistic effects of the cotreatment with a relatively low-concentration GLE and QCT or ISL. Relatively low and high-concentrations were defined based on our previous studies: 100 mg/kg for *Cordyceps militaris* extracts [20], 30 mg/kg quercetin and isoliquiritigenin [33]. The mice in the group receiving a relatively high-concentration cotreatment were orally administrated drinking water, GLE (30 mg/kg), QCT (30 mg/kg), ISL (30 mg/kg), GLE + QCT (30 mg/kg + 30 mg/kg), or GLE + ISL (30 mg/kg + 30 mg/kg) for 23 days. The mice in the group receiving the low-concentration cotreatment were orally administrated drinking water, GLE (10 mg/kg), QCT (10 mg/kg), ISL (10 mg/kg), GLE + QCT (10 mg/kg + 10 mg/kg), or GLE + ISL (10 mg/kg + 10 mg/kg) for 23 days. The tumors were identified and tumor size was measured every other day using a standard caliper and calculated using the following equation: tumor size = [tumor length (mm) × tumor width (mm)^2^]/2, as previously described [20]. After the tumor size reached approximately 1000 mm^3^, the mice were euthanized by an overdose of isoflurane and the euthanasia was confirmed by cervical dislocation. Then, the tumors were harvested surgically.

### 4.5. Cytotoxicity Assay

The cytotoxic effects of GLE and GAA in SNU719 cells were evaluated using a cytotoxicity assay performed using the cell counting kit-8 (CCK-8; Dojindo, Kumamoto, Japan), according to the manufacturer’s protocol. Briefly, 100 μL of cell suspension (1 × 10^4^ cells/well) was seeded into a 96-well plate. On the following day, GLE or GAA was applied at various concentrations (0 ~ 2000 mg/mL for GLE and 0 ~ 5000 μM for GAA). In 48 h post treatment, 10 μL of CCK-8 solution was added to each sample. The samples were incubated for another 3 h and then the absorbance of each cell suspension was measured at 450 nm using an enzyme-linked immunosorbent assay reader. The cytotoxic effects of combined GLE and QCT in SNU719 cells were also evaluated using the CCK-8 cytotoxicity assay mentioned above. Briefly, 100 μL of cell suspension (1 × 10^4^ cells/well) was seeded into a 96-well plate. On the following day, the combinations of GLE and QCT were made in a series of concentrations (0.000 ~ 0.053 mg/mL for GLE, 0.00 ~ 124.00 μM for QCT) and these combinations were added to the cell suspension in a 96-well plate. Further, 10 μL of CCK-8 solution was added to each sample in 48 h post treatment and the cytotoxicity of the combined compounds were calculated as 450 nm absorbance values of cell suspension. The relative inhibition (RI) of the sample treatment (ST) was evaluated by subtracting RI of the negative control treatment (CT); RI value for ST = 450 nm absorbance value of CT–absorbance value of ST. The measurement of RI values were conducted in triplicate. A synergistic effect was evaluated based on the Chou-Talalay analysis [19]. First, the intercepts and the slope were assessed from a dose response curve with RI values from GLE and QCT treatments. The RI was used to evaluate the combination index (CI) with a combination of GLE and QCT. The GLE treatments made three data points (0.01 ~ 0.05 mg/mL) and the QCT treatments made 10 data points (0.02 ~ 124.00 μM). Subsequently, both the Fa-CI plot (Fa = fraction affected) and median-effect plot were employed to determine whether the interactions between the two compounds were additive, synergistic or antagonistic. This method involved plotting the dose-effect equation: fa/fu = (D/Dm)m(1)

In Equation (1), D is the dose, Dm is the required dose for 50% inhibition of cell growth, fa is the faction affected by dose D, fu is the unaffected fraction and m is a coefficient of the sigmoidicity of the dose-effect curve. Both the Fa-CI plot and median-effect plot yielded similar conclusions of synergism or antagonism. Synergism, additivism or antagonism of the combined effects were quantitatively represented by CI. CI < 1.0, 1.0 < CI < 1.1, and CI > 1.1 indicate synergism, additivism and antagonism, respectively.

### 4.6. Cell Viability Assay

The effects of GLE, QCT, and GAA on SNU719 cell viability were determined using MUSE count and viability kit (Merck Millipore, Darmstadt, Germany), according to the manufacturer’s protocol. The cells (2 × 10^6^) were seeded into a 6-well plate and treated for 48 h with mixtures of GLE and QCT or GLE and GAA at various concentrations (0 ~ 0.027 mg/mL for GLE, 0 ~ 7.75 μM for QCT, and 0 ~ 0.49 mM for GAA). A negative control sample was treated with the same volume of DMSO for 48 h. The treated cells were trypsinized, harvested and resuspended as 10^6^ ~ 10^7^ cells/mL of fresh serum. Cell suspension (20 μL) was mixed with 380 μL of the MUSE count and viability reagent and then incubated at 25 °C for 5 min. Cell viability was measured using MUSE cell analyzer (Millipore, Burlington, MA, USA).

### 4.7. Apoptosis Assay

The apoptosis effects of QCT and GAA on SNU719 cells were determined using MUSE Annexin V & Dead cell assay kit (Merck Millipore, Darmstadt, Germany), according to the manufacturer’s protocol. The cells (2 × 10^6^) were seeded into a 6-well plate and treated for 48 h with mixtures of QCT and GAA at various concentrations (0 and 0.96 μM for QCT, and 0 and 0.49 mM for GAA). A negative control sample was treated with the same volume of DMSO for 48 h. The treated cells were trypsinized, harvested and resuspended as 10^5^ ~ 10^6^ cells/mL of fresh serum. Cell suspension (180 μL) was mixed with 20 μL of the MUSE Annexin V & Dead Cell reagent and then incubated at 25 °C for 20 min. Cell viability was measured using MUSE cell analyzer (Millipore, Burlington, MA, USA).

### 4.8. Luciferase Assay

SNU719 cells were transfected using the Neon system (Invitrogen, Carlsbad, CA, USA) with a luciferase reporter constructs that contained a Renilla luciferase gene transcribed by the promoter of EBV lytic gene, BHLF1. Transfected SNU719 cells were then selected with blasticidin (Invitrogen) for approximately 30 days and established as SNU719-BHLF1 luciferase cells for the luciferase assay. SNU719-BHLF1 luciferase cells were cotreated for 48 h at 37 °C with mixtures of GLE and QCT or GLE and GAA at various concentrations (0 ~ 0.027 mg/mL for GLE, 0 ~ 7.75 μM for QCT, and 0 ~ 0.49 mM for GAA). Luciferase activity was measured using a dual-luciferase reporter assay system (Promega, Madison, WI, USA), according to the manufacturer’s protocol.

### 4.9. Western Blot Assay

To assess the proapoptotic effects of the cotreatment with GLE and QCT, a western blot analysis was performed using SNU719 cells cotreated with GLE and QCT. The total proteins were harvested using trypsin for 48 h. Cells (10^7^) were lysed using 100 μL of a reporter lysis buffer (Promega, Madison, WI) supplemented with proteinase inhibitor and phenylmethylsulfonyl fluoride. The lysates were further fractionated using a bioruptorsonicator (5 min, 30 s on/off pulses). The total protein content in cell lysates was measured using the Bradford assay. Equivalent amounts of total proteins were subjected to a western blot analysis. The whole proteins were separated using 10% sodium dodecyl sulfate (SDS)-polyacrylamide gel electrophoresis and transferred to membranes. In addition, cytoplasmic and nuclear proteins of the treated SNU719 cells were recovered using NE-PER nuclear and cytoplasmic extraction reagents (Thermo Scientific, Waltham, MA, USA). The cytoplasmic and nuclear protein expression was determined using a western blot analysis. The membranes were probed with antibodies against the proteins of interest. Anti-poly [ADP-ribose] polymerase 1 (PARP1), anti-Bax, anti-caspase 3, anti-cytochrome C, anti-Bcl-2, anti-glyceraldehyde 3-phosphate dehydrogenase (GAPDH), and anti-proliferating cell nuclear antigen (PCNA) antibodies were purchased from Cell Signaling Technology (Danvers, MA, USA) and used to detect corresponding proteins. The expression patterns of GAPDH and PCNA were used as internal controls. The antibody-bound proteins were visualized by enhanced chemiluminescence (ECL) detection reagent (GE Healthcare, Chicago, IL, USA). The membranes were stripped and reprobed with other antibodies.

### 4.10. Quantitative Reverse-Transcriptase PCR Assay

To assess the effects of the cotreatment with GLE and QCT on gene expression, a qRT-PCR assay was performed using SNU719 cells cotreated with GLE and QCT. RNA was isolated from the cells using the RNeasy mini kit (Qiagen, Germantown, MD, USA). Purified RNA was converted into cDNA by using superscript III reverse transcriptase (Invitrogen, Carlsbad, CA, USA). The primers specific for EBV genes were used. The sequences of primer sets used in this study can be provided upon request.

### 4.11. LC-MS/MS Analysis

The GLE samples and GAA standard were analyzed using a Shimadzu LCMS-8060 liquid chromatograph-mass spectrometer system (Shimadzu, Tokyo, Japan) equipped with an electrospray ionization (ESI) interface. Analyte separation was performed using the Kinetex C18 column (100 × 2.10 mm, 2.6 μm, 100 Å; Phenomenex, Torrance, CA, USA). The mobile phase consisted of 0.1% formic acid in water (A) and 0.1% formic acid in acetonitrile (B) and was run on the following gradient: 0–1 min (20% B–40% B), 1–6 min (50% B), and 6.1–10 min (20% B). The total run time was 9 min and the flow rate was 0.2 mL/min. The total run time was 10 min and the flow rate was 0.2 mL/min. Electrospray ionization was performed in negative-ion mode at −3500 V. The optimum operating conditions were determined as follows: capillary temperature, 350 °C; vaporizer temperature, 300 °C; collision gas (argon) pressure, 1.5 mTorr. The quantitation was conducted in multiple reaction monitoring (MRM) mode at 515.3 > 285.3 and the collision energy (CE) value was 40 eV.

### 4.12. Statistical Analysis

The *p*-values were calculated by a 2-tailed Student’s *t*-test using Excel (Microsoft, Redmond, WA, USA). In addition, all statistical analyses were subjected to a one-way analysis of variance (ANOVA) and re-verified by Turkey’s multiple comparison test using GraphPad Prism (San Diego, CA, USA).

## Figures and Tables

**Figure 1 molecules-24-03834-f001:**
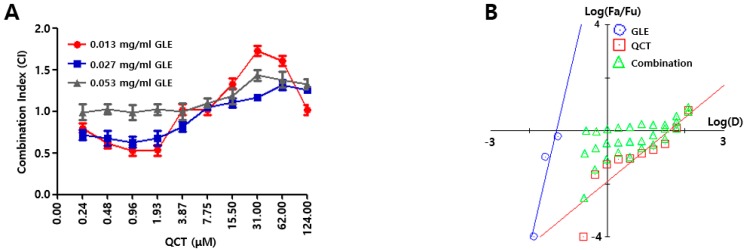
Synergism of GLE and QCT in causing cytotoxicity. The measurements of cytotoxic synergism of the combinations of GLE and QCT in SNU719 cells using Fa-CI plot (**A**) and the median-effect plot (**B**). Combinations of GLE and QCT were made in a series of concentrations and applied to the CCK8. GLE, QCT and CI: *Ganoderma lucidum* extracts, quercetin and combination index, respectively.

**Figure 2 molecules-24-03834-f002:**
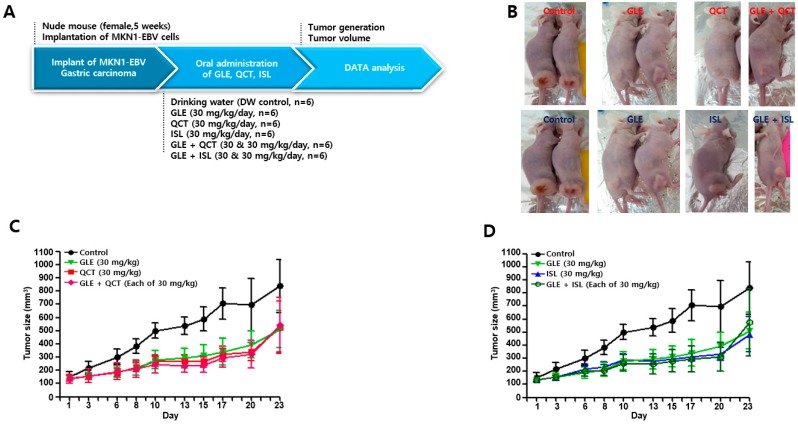
Cotreatment with GLE and QCT at relatively high concentration did not synergistically suppress EBVaGC development in xenograft mouse model. (**A**) A schematic diagram of the antitumor assay carried out using xenograft mouse model. (**B**) A representative image of a nude mouse bearing MKN1-EBV-derived tumors. (**C**) The determination of the antitumor effects of relatively high-concentration GLE (30 mg/kg), QCT (30 mg/kg) and mixture of GLE and QCT (30 + 30 mg/kg) in MKN1-EBV-derived tumors in nude mice. (**D**) The determination of the antitumor effects of high-concentration GLE (30 mg/kg), ISL (30 mg/kg) and a mixture of GLE and ISL (30 + 30 mg/kg) in MKN1-EBV-derived tumors in nude mice. Any group was not statistically different in tumor sizes. GLE, QCT and ISL: *Ganoderma lucidum* extracts, quercetin, and isoliquiritigenin, respectively.

**Figure 3 molecules-24-03834-f003:**
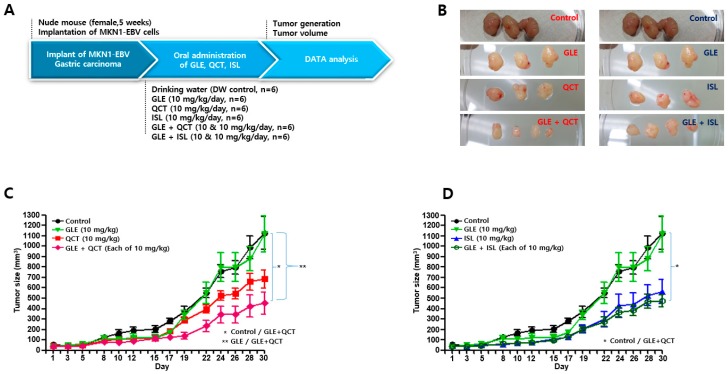
Cotreatment with GLE and QCT at relatively low concentration synergistically suppressed EBVaGC development in xenograft mouse model. (**A**) A schematic diagram of the antitumor assay carried out using a xenograft mouse model. (**B**) A representative image of MKN1-EBV-derived tumors recovered form xenograft nude mice. (**C**) The determination of the antitumor effects of relatively low-concentration GLE (10 mg/kg), QCT (10 mg/kg), and mixture of GLE and QCT (10 + 10 mg/kg) in MKN1-EBV-derived tumors in nude mice. (**D**) The determination of the antitumor effects of low-concentration GLE (10 mg/kg), ISL (10 mg/kg) and a mixture of GLE and ISL (10 + 10 mg/kg) in MKN1-EBV-derived tumors in nude mice. Control versus GLE + QCT, GLE versus GEL + QCT and control versus GLE + ISL were statistically different in tumor sizes. GLE, QCT, and ISL: *Ganoderma lucidum* extracts, quercetin, and isoliquiritigenin, respectively.

**Figure 4 molecules-24-03834-f004:**
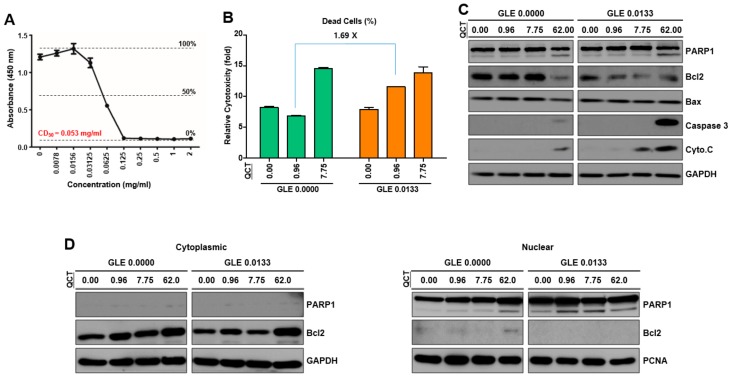
Cotreatment with GLE and QCT decreased cell viability and induced apoptosis. (**A**) Measurement of CD_50_ of GLE in SNU719 cells. GLE CD_50_ was 0.0532 mg/mL. (**B**) The determination of the cytotoxic effects of cotreatment with 0.0133 mg/mL GLE and various concentrations of QCT in SNU719 cells. (**C**) A western blot analysis for the determination of the expression of apoptotic proteins in total proteins derived from SNU719 cells. (**D**) A western blot analysis for the determination of the expression of apoptotic proteins in cytoplasmic and nuclear fractions of SNU719 cells. GLE and QCT: *Ganoderma lucidum* extracts and quercetin, respectively.

**Figure 5 molecules-24-03834-f005:**
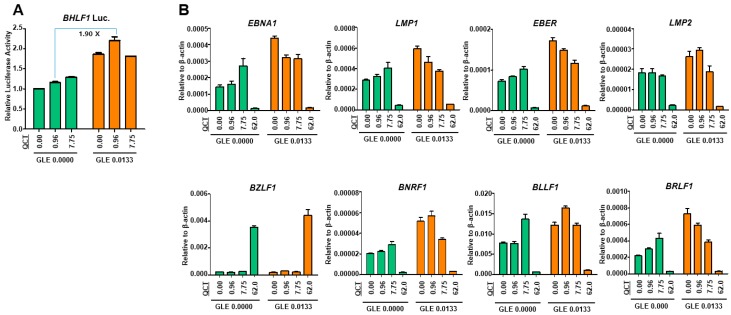
Cotreatment with GLE and QCT induced EBV lytic reactivation and gene expression. Luciferase assay was conducted to determine EBV lytic reactivation in SNU719-BHLF1 luciferase cells. Relative luciferase activity measured in the SNU719-BHLF1 luciferase cells. (**A**) SNU719-BHLF1 luciferase cells were treated with low-concentration GLE (0.0133 mg/mL) and various concentrations of QCT to evaluate their stimulatory effects on EBV lytic reactivation. SNU719 cells were treated with both various concentrations of QCT and 0.0133 mg/mL GLE. (**B**) The determination of the effects of various concentrations of QCT + GLE (0.0133 mg/mL) on the EBV gene expression. GLE and QCT: *Ganoderma lucidum* extracts and quercetin, respectively.

**Figure 6 molecules-24-03834-f006:**
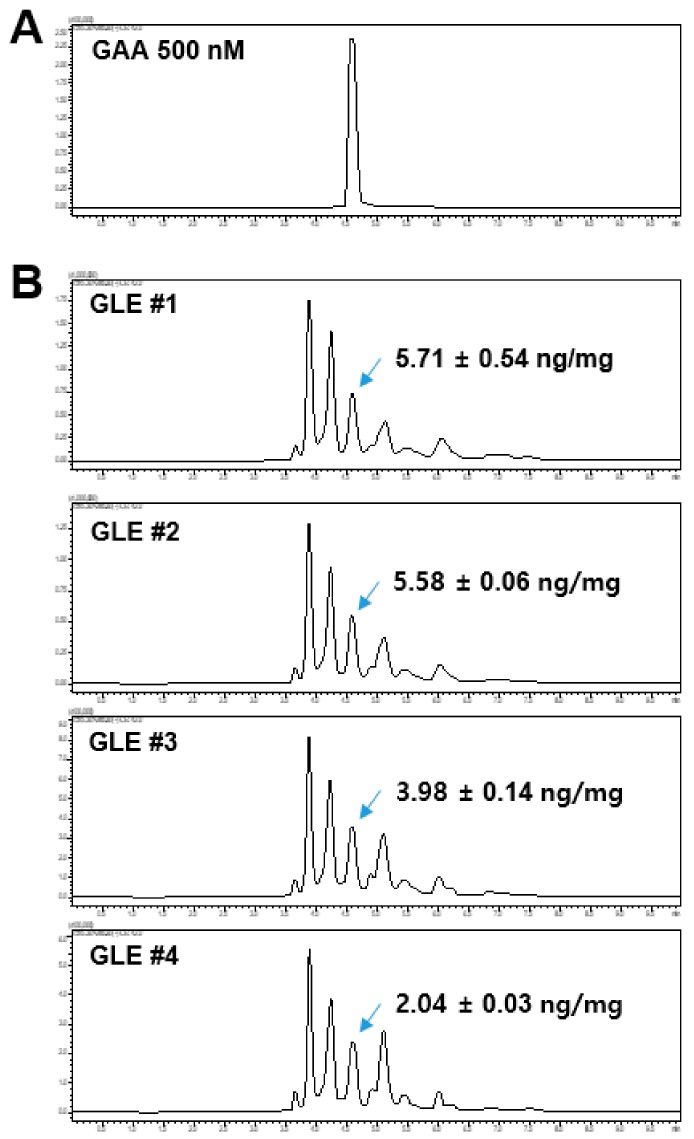
Identification of GAA in GLE. (**A**) LC-MS/MS profile of 500 nM GAA. (**B**) LC-MS/MS profiles of GLE samples. GAA and GLE: Ganoderic acid A and *Ganoderma lucidum* extracts, respectively.

**Figure 7 molecules-24-03834-f007:**
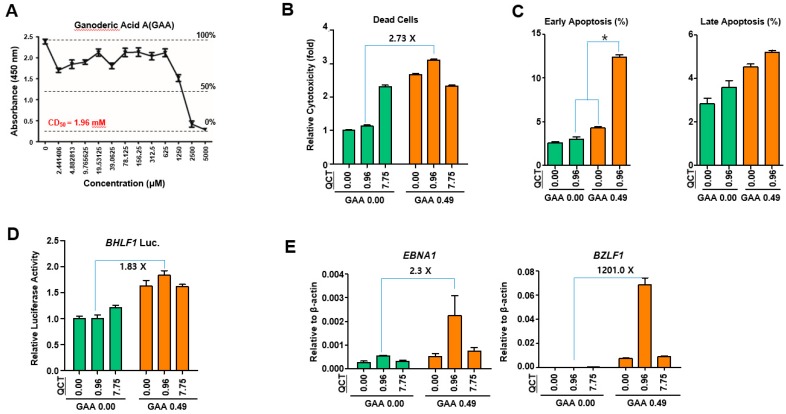
Cotreatment with GAA and QCT decreased cell viability. (**A**) The measurement of the CD_50_ of GAA in SNU719 cells. The CD_50_ value of GAA was 1.96 mM. (**B**) The determination of the cytotoxic effects of the cotreatment with 0.49 mM GAA and various concentrations (0 ~ 7.75 μM) of QCT in SNU719 cells. (**C**) The determination of the apoptotic effects of the cotreatment of 0.49 mM GAA and 0.96 μM QCT. (**D**) The determination of the effects of the cotreatment with GLE and QCT on the EBV lytic reactivation. SNU719-BHLF1 luciferase cells were treated with 0.49 mM GAA and various concentrations (0 ~ 7.75 μM) of QCT. (**E**) The determination of the effects of various concentrations of QCT + GAA (0.49 mM) on EBV gene expression. GAA and QCT: Ganoderic acid A and quercetin, respectively.

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
