# Peer review of "Quercetin Synergistically Inhibit EBV-Associated Gastric Carcinoma with *Ganoderma lucidum* Extracts"

_molecules, 2019, doi:10.3390/molecules24213834_

Round 1

Reviewer 1 Report

Hur et al present an interesting account of studies indicating the inhibition of EBV-associated gastric carcinoma by synergistic effect of quercetin (QCT) with Ganoderma lucidum extracts (GLE). They show that co-treatment of GLE and QCT at low concentration suppressed the tumor growth to a significant level. Interestingly, the authors are trying to come up with the mechanism of action of tumor inhibition is due by synergistic effects of the injected compounds(/extract) to induce apoptosis and lytic reactivation. They conclude that there is a synergistic role of QCT supplemented with GLE in the effective treatment of EBVaGC and this is due to activating apoptosis in the indicated cells.

While interesting, the paper is a little speculative and there may be some more direct ways of drawing conclusions on the action of QCT and GLE. The following points should be carefully considered by the authors in order to try to improve paper and make it a little more rigorous and conclusive.

Major points:

In the first part of the paper the authors conclude that the combination of QCT and GLE synergistically affect the tumor growth and tumor cell viability. If this is the case, the authors have not provided enough evidence of the constituents of GLE. I believe it is very important to know the constituents of the extract (GLE). Simple biochemical assays will reveal basic bio-constituents present in the extract. The authors show high dose of the combination of compound and extract is not good to inhibit the tumor (Fig 2). I am not convinced with the statistical proof they are providing. The authors should do ANOVA first followed by Posthoc Tukey’s test to clearly tell which part of the data is contributing to significance of the result. This is followed by Posthoc students t-test between the groups at each day to tell exactly from which day onwards the tumor started to decrease. Fig 4: Authors are concluding the cause of reduction of tumor in GLE and QCT injected animals is due to induction of apoptosis. Although the authors are making a very good point to come up with the action of drugs, but still the evidence of apoptosis is lacking. There is no proof that the cells are undergoing apoptosis not necrosis. Just by showing percentage of cell death and PARP1 and Bcl2 blots is not convincing of apoptosis. Recent studies have also been shown that cleaved PARP1 found with the necrosis (Douglas and Baines., 2014; Shin et al., 2015). 5: The authors are trying to prove their point of apoptosis induction could be due to lytic reactivation. Basically, lytic reactivation can be occurred in any inflammatory response. So, without giving enough evidence of apoptosis there will be no point of considering lytic reactivation data.

Minor points:

Result heading 2.2 and 2.3 is confusing. High concentration (2.2) and low concentration (2.3) both are suppressing tumor growth.

References;

Douglas DL, Baines CP. PARP1-mediated necrosis is dependent on parallel JNK and Ca2+/calpain pathways. J Cell Sci. 2014 Oct 1;127(19):4134-45.

Shin HJ, Kwon HK, Lee JH, Gui X, Achek A, Kim JH, Choi S. Doxorubicin-induced necrosis is mediated by poly-(ADP-ribose) polymerase 1 (PARP1) but is independent of p53. Scientific reports. 2015 Nov 2;5:15798.

Author Response

Reviewer #1’s comments

Hur et al present an interesting account of studies indicating the inhibition of EBV-associated gastric carcinoma by synergistic effect of quercetin (QCT) with Ganoderma lucidum extracts (GLE). They show that co-treatment of GLE and QCT at low concentration suppressed the tumor growth to a significant level. Interestingly, the authors are trying to come up with the mechanism of action of tumor inhibition is due by synergistic effects of the injected compounds(/extract) to induce apoptosis and lytic reactivation. They conclude that there is a synergistic role of QCT supplemented with GLE in the effective treatment of EBVaGC and this is due to activating apoptosis in the indicated cells.

While interesting, the paper is a little speculative and there may be some more direct ways of drawing conclusions on the action of QCT and GLE. The following points should be carefully considered by the authors in order to try to improve paper and make it a little more rigorous and conclusive.

Major points:

In the first part of the paper the authors conclude that the combination of QCT and GLE synergistically affect the tumor growth and tumor cell viability. If this is the case, the authors have not provided enough evidence of the constituents of GLE. I believe it is very important to know the constituents of the extract (GLE) (Comment #1-1). Simple biochemical assays will reveal basic bio-constituents present in the extract.

Response to comment #1-1: We conducted HPLC assay with GLE samples and found that GAAs were included in GLE samples used in this study. This result was added as Figure 6 in manuscript. 

The authors show high dose of the combination of compound and extract is not good to inhibit the tumor (Fig 2). I am not convinced with the statistical proof they are providing. The authors should do ANOVA first followed by Posthoc Tukey’s test to clearly tell which part of the data is contributing to significance of the result (Comment #1-2). This is followed by Posthoc students t-test between the groups at each day to tell exactly from which day onwards the tumor started to decrease.

Response to comment #1-2: We statistically reanalyzed our animal study with ANONA and Turkey’s Multiple Comparison Test using GraphPad Prism. Previous figures (Fig. 2C, Fig. 2D, Fig. 3C, Fig. 3D) were replaced with these reanalyzed figures.

Fig 4: Authors are concluding the cause of reduction of tumor in GLE and QCT injected animals is due to induction of apoptosis. Although the authors are making a very good point to come up with the action of drugs, but still the evidence of apoptosis is lacking. There is no proof that the cells are undergoing apoptosis not necrosis. Just by showing percentage of cell death and PARP1 and Bcl2 blots is not convincing of apoptosis. Recent studies have also been shown that cleaved PARP1 found with the necrosis (Douglas and Baines., 2014; Shin et al., 2015). 5: The authors are trying to prove their point of apoptosis induction could be due to lytic reactivation (Comment #1-3). Basically, lytic reactivation can be occurred in any inflammatory response. So, without giving enough evidence of apoptosis there will be no point of considering lytic reactivation data.

Response to comment #1-3: We conducted another western blot analysis with whole proteins of SNU719 cells treated GLE and QCT. This analysis showed that Caspase 3 and Cytochrome C were also upregulated by cotreatment of GLE and QCT. In contrast, Bcl2, anti-apoptotic factor was downregulated by the GLE and QCT cotreatment. Thus, Along with PARP1, regulation of apoptotic and anti-apoptotic factors was indicative of apoptosis induction by the GLE and QCT cotreatment. We added the new data as Fig. 4C in manuscript.    

Minor points:

Result heading 2.2 and 2.3 is confusing. High concentration (2.2) and low concentration (2.3) both are suppressing tumor growth (Comment #1-4).

Response to comment #1-4: We retitled as “Cotreatment of GLE and QCT at high concentration did not synergistically cause anti-tumor effect” and “Cotreatment of GLE and QCT at high concentration synergistically caused anti-tumor effect”. 

References;

Douglas DL, Baines CP. PARP1-mediated necrosis is dependent on parallel JNK and Ca2+/calpain pathways. J Cell Sci. 2014 Oct 1;127(19):4134-45.

Shin HJ, Kwon HK, Lee JH, Gui X, Achek A, Kim JH, Choi S. Doxorubicin-induced necrosis is mediated by poly-(ADP-ribose) polymerase 1 (PARP1) but is independent of p53. Scientific reports. 2015 Nov 2;5:15798.

Reviewer 2 Report

Manuscript ID: molecules-608720

Quercetin synergistically inhibit EBV-associated gastric carcinoma with Ganoderma lucidum extracts

In the manuscript " Quercetin synergistically inhibit EBV-associated gastric carcinoma with Ganoderma lucidum extracts  " by Sora Hur , et al., aimed to investigate G. lucidum extracts as potential antitumor agents and a counterpart of quercetin (QCT) for
cotreatment in suppressing epstein-Barr virus (EBV)-associated gastric carcinoma (EBVaGC) development. Therefore, antitumor assays using EBVaGC xenograft mice model were performed and the promising obtained results showed that GLE could suppress tumor development. In addition the authors showed that these inhibitory effects was significantly increased by low concentration of quercetin (QCT) cotreatment in the xenograft mice. They also observed that GLE in low concentrations synergistically reinforced QCT-induced apoptosis and EBV lytic reactivation. The obtained results suggestingf that QCT-
supplemented GLE could be a potential food adjunct for prevention of EBVaGC
development are very interesting for the scientific community.

The paper fit the aims and scope of the Molecules. The title is informative and give a clear idea of the paper.

The work is very interesting, technically sound and scientifically valid. Is of great interest for Molecules readers and for the scientific community, in general. The used methodology is appropriate to the objectives of the work and the experimental design was properly conducted and described. In addition, the conclusions drawn are fully supported by the data presented and the claims are appropriately discussed in the context of previous literature. In addition, the paper provided sufficient methodological detail that the experiments can be reproduced.

The Abstract is adjusted to the developed work, to the used methodologies and to the obtained results. A correction is need at line 38-39 as follow “…. The aim of this study was to investigate GLE as potential antitumor agentes …”

Besides large, the INTRODUCTION is interesting, describing the current state of art of the subject in addition to discussion in the context of previous literature including some important published works.

The figures needs some rearrangements. Near the figures the word “Figure X.” x=1, 2, …,  is in duplicate and deserves correction.

On DISCUSSION section the number of diferente sections from 1,5 to 1.11 needs correction:  1.5.Cytotoxicity assay; 1.6.Cell viability assay ; 1.7.Apoptosis assay; 1.8.Luciferase assay; 1.9.Western blot assay ; 1.10. Quantitative reverse-transcription PCR assay; and 1.11. Statistical analysis, must be corrected as follow :

4.5.Cytotoxicity assay

4.6.Cell viability assay

4.7.Apoptosis assay;

4.8.Luciferase assay;

4.9.Western blot assay ;

4.10. Quantitative reverse-transcription PCR assay;

4.11 Statistical analysis

Along all manuscript a point is need close to figure abbreviation (Fig). Should be Fig. 6, for instance, and not Fig 6

Nice study, very well planed and designed, very well structured and discussed, with interesting results. Congratulations!

Author Response

Reviewer #2’s comments

In the manuscript " Quercetin synergistically inhibit EBV-associated gastric carcinoma with Ganoderma lucidum extracts  " by Sora Hur , et al., aimed to investigate G. lucidum extracts as potential antitumor agents and a counterpart of quercetin (QCT) for
cotreatment in suppressing epstein-Barr virus (EBV)-associated gastric carcinoma (EBVaGC) development. Therefore, antitumor assays using EBVaGC xenograft mice model were performed and the promising obtained results showed that GLE could suppress tumor development. In addition the authors showed that these inhibitory effects was significantly increased by low concentration of quercetin (QCT) cotreatment in the xenograft mice. They also observed that GLE in low concentrations synergistically reinforced QCT-induced apoptosis and EBV lytic reactivation. The obtained results suggestingf that QCT-
supplemented GLE could be a potential food adjunct for prevention of EBVaGC
development are very interesting for the scientific community.

The paper fit the aims and scope of the Molecules. The title is informative and give a clear idea of the paper.

The work is very interesting, technically sound and scientifically valid. Is of great interest for Molecules readers and for the scientific community, in general (Comment #2-1). The used methodology is appropriate to the objectives of the work and the experimental design was properly conducted and described. In addition, the conclusions drawn are fully supported by the data presented and the claims are appropriately discussed in the context of previous literature. In addition, the paper provided sufficient methodological detail that the experiments can be reproduced.

Response to comment #2-1:  We so much appreciated reviwer’s this encouraging comment.

The Abstract is adjusted to the developed work, to the used methodologies and to the obtained results. A correction is need at line 38-39 as follow “…. The aim of this study was to investigate GLE as potential antitumor agentes …” (Comment #2-2)

Response to comment #2-2: We corrected as reviewer suggested.

Besides large, the INTRODUCTION is interesting, describing the current state of art of the subject in addition to discussion in the context of previous literature including some important published works.

The figures needs some rearrangements. Near the figures the word “Figure X.” x=1, 2, …,  is in duplicate and deserves correction (Comment #2-3).

Response to comment #2-3: We corrected as reviewer suggested.

On DISCUSSION section the number of diferente sections from 1,5 to 1.11 needs correction:  1.5.Cytotoxicity assay; 1.6.Cell viability assay ; 1.7.Apoptosis assay; 1.8.Luciferase assay; 1.9.Western blot assay ; 1.10. Quantitative reverse-transcription PCR assay; and 1.11. Statistical analysis, must be corrected as follow (Comment #2-4) :

4.5.Cytotoxicity assay

4.6.Cell viability assay

4.7.Apoptosis assay;

4.8.Luciferase assay;

4.9.Western blot assay ;

4.10. Quantitative reverse-transcription PCR assay;

4.11 Statistical analysis

Response to comment #2-4: We corrected as reviewer suggested.

Along all manuscript a point is need close to figure abbreviation (Fig). Should be Fig. 6, for instance, and not Fig 6 (Comment #2-5)

Response to comment #2-5: We corrected as reviewer suggested.

Nice study, very well planed and designed, very well structured and discussed, with interesting results. Congratulations! (Comment #2-6)

Response to comment #2-6:  We so much appreciated reviwer’s this encouraging comment.

Reviewer 3 Report

In this manuscript, the authors report for the first time a study on the synergistic effect of quercetin (QCT) in combination with Ganoderma lucidum Extract (GLE) as potential antitumor therapy. A considerable amount of experimental work is presented, and promising results are reported; antitumor assays using EBVaGC xenograft mice model showed that tumor growth suppression by GLE was significantly augmented by low concentration of QCT. The results suggests that QCT-supplemented GLE could be used as a food supplement with potential for prevention of EBVaGC development.

Thus, this manuscript is worth of publication on Molecules, provided that a careful revision of the English language would be carried out by a mother-tongue revisor.

Author Response

Reviewer #3’s comments

In this manuscript, the authors report for the first time a study on the synergistic effect of quercetin (QCT) in combination with Ganoderma lucidum Extract (GLE) as potential antitumor therapy. A considerable amount of experimental work is presented, and promising results are reported; antitumor assays using EBVaGC xenograft mice model showed that tumor growth suppression by GLE was significantly augmented by low concentration of QCT. The results suggests that QCT-supplemented GLE could be used as a food supplement with potential for prevention of EBVaGC development.

Thus, this manuscript is worth of publication on Molecules (Comment #3-1), provided that a careful revision of the English language would be carried out by a mother-tongue revisor (Comment #3-2).

Response to comment #3-1: We so much appreciated reviwer’s this encouraging comment.

Response to comment #3-2: We revised our manuscript with several native English speakers and fixed some English errors in manuscript.